# Post-acute sequelae of COVID-19 among hospitalized patients in Estonia: Nationwide matched cohort study

Anna Tisler[1]*, Oliver Stirrup[2], Heti Pisarev[1], Ruth Kalda[1], Tatjana Meister[1], Kadri Suija[1], Raivo Kolde[3], Marko Piirsoo[4], Anneli Uusküla[1]

1 Institute of Family Medicine and Public Health, University of Tartu, Tartu, Estonia, 2 Institute for Global Health, University College London, London, United Kingdom, 3 Institute of Computer Science, University of Tartu, Tartu, Estonia, 4 Institute of Technology, University of Tartu, Tartu, Estonia

* anna.tisler@ut.ee

## Abstract

### Background

Post-acute COVID-19 sequelae refers to a variety of health complications involving different organ systems that have been described among individuals after acute phase of illness. Data from unselected population groups with long-time follow up is needed to comprehensively describe the full spectrum of post-acute COVID-19 complications.

### Methods

In this retrospective nationwide cohort study, we used data obtained from electronic health record database. Our primary cohort were adults hospitalized with confirmed COVID-19 and matched (age, sex, Charlson Comorbidity Index) unaffected controls from general population. Individuals included from February 2020 until March 2021 were followed up for 12 months. We estimated risks of all-cause mortality, readmission and incidence of 16 clinical sequelae after acute COVID-19 phase. Using a frailty Cox model, we compared incidences of outcomes in two cohorts.

### Results

The cohort comprised 3949 patients older than 18 years who were alive 30 days after COVID-19 hospital admission and 15511 controls. Among cases 40.3% developed at least one incident clinical sequelae after the acute phase of SARS-CoV-2 infection, which was two times higher than in general population group. We report substantially higher risk of all-cause mortality (adjusted hazard ratio (aHR) = 2.57 (95%CI 2.23–2.96) and hospital readmission aHR = 1.73 (95%CI 1.58–1.90) among hospitalized COVID-19 patients. We found that the risks for new clinical sequalae were significantly higher in COVID-19 patients than their controls, especially for dementia aHR = 4.50 (95% CI 2.35–8.64), chronic lower respiratory disease aHR = 4.39 (95% CI 3.09–6.22), liver disease aHR 4.20 (95% CI 2.01–8.77) and other (than ischemic) forms of heart diseases aHR = 3.39 (95%CI 2.58–4.44).

**Data Availability Statement:** All relevant data are within the paper and its Supporting Information files.

**Funding:** Research was carried out with the support of Estonian Research Council. The funder had no role in study design, data collection and analysis, decision to publish, or preparation of the manuscript. Grants: PRG1197, PRG198.

**Competing interests:** The authors have declared that no competing interests exist.

## Conclusion

Our results provide evidence that the post-acute COVID-19 morbidity within the first year after COVID-19 hospitalization is substantial. Risks of all-cause mortality, hospitalisation and majority of clinical sequelae were significantly higher in hospitalized COVID-19 patients than in general population controls and warrant targeted prevention efforts.

## Introduction

SARS-CoV-2 (the cause of COVID-19) leads to high levels of morbidity and mortality among unvaccinated individuals, and the virus created a novel public health emergency with substantial burden on healthcare systems [1]. Since emerging in 2020 the virus has caused more than 500 million COVID-19 cases and 6 million deaths globally with one third reported from Europe [2]. More severe cases of COVID-19 and higher mortality were reported among older individuals, male, obese and those with comorbidities [3]. There is an increasing body of evidence reporting that COVID-19 patients are at substantially higher risk of death and hospital readmission and also continue to suffer from complications including persistent severe symptoms and organ dysfunction.

Post-acute COVID-19 syndrome is characterized by persistent symptoms and/or delayed or long-term complications beyond 4 weeks after acute phase [4]. Acute COVID-19 can potentially lead to new clinical events or exacerbate pre-existing (chronic) conditions. A study on an adult population in UK documented that older individuals, those having pre-existing medical conditions, and those admitted to hospital because of acute COVID-19 were at greatest excess risk for clinical sequelae after the acute phase of SARS-CoV-2 within the subsequent 6-month period [5]. A systematic review has shown that 10.3% of COVID-19 survivors required hospital readmission within one-year post-discharge, and the one-year post-discharge all-cause mortality rate of COVID-19 patients was 7.9% [6]. Hospital readmissions are a concern for public health as they impose additional burden on the healthcare system and demand more resources.

A large study including 273,618 COVID-19 survivors reported that there is association between severity of COVID-19 and incidence of long-COVID consequences: patients with more severe illness (hospitalization, intensive care unit (ICU) admission) had significantly more long-term effect overall [7]. It is critical that we describe and assess the likelihood of post-acute COVID-19 sequelae; also quantify / compare the consequences of COVID-19 survivors to that in comparable (age, gender, comorbidities) profile control groups, that allow the inference of counterfactual outcomes in order to be able to provide appropriate care and tailor treatment for those complications that are most often seen among patients. To address this need and generate new knowledge in this important area, we aimed to investigate the post-acute COVID-19 sequelae among hospitalized COVID-19 individuals up to 12 months to determine the incidence of all-cause mortality, readmissions and characteristics of new onset diseases following severe COVID-19 in Estonia.

## Materials and methods

### Data source

The data for this nationwide retrospective study was obtained from the Estonian Health Insurance Fund (EHIF), Estonian Health Board (EHB) and Estonian Causes of Death Registry

(ECDR). The EHIF, EHB and ECDR are all population-based and nation-wide health/administration organizations. Unique personal identification codes which are assigned to all Estonian residents at birth or at the time of immigration were used for data linkage. Confirmed COVID-19 cases were those based on real-time polymerase chain reaction (PCR) testing on nasopharyngeal specimens by an accredited molecular diagnostics laboratory using certified methods [8]. Data on the date of SARS-CoV-2 RNA test and results were obtained from EHB. Personal information on gender and date of birth and health care utilization with dates of service, diagnostic codes based on International Statistical Classification of Diseases and Related Health Problems 10th Revision (ICD-10), were obtained from EHIF. As of December 2021, the EHIF contained information on 1 265 601 individuals or 94% of the Estonian population with insurance coverage.

## Population and setting

We included data on all people aged 18 and over who tested positive for SARS-CoV-2 during the first year of the COVID-19 epidemic (26 February 2020 to 28 February 2021) and were hospitalized during the acute phase of COVID-19. For the current study we refer to the date of the first positive SARS-CoV-2 test as the index date. Individuals with COVID-19 entered the cohort on the index date. In our study we focused on the pre-vaccination period of the pandemic, so we included individuals with the index dates before mass vaccination in February 2021. Participants with COVID-19 who were admitted to a hospital within 30 days after or 7 days before their positive SARS-CoV-2 test were considered as cases. Those cases who died during acute phase of disease meaning during 30 days of the hospitalization were excluded.

Using EHIF listing as a matching source up to four controls for each case were selected. Eligible for matching were those without evidence of SARS-CoV-2 infection at the index date of each case, were alive and without evidence of SARS-CoV-2 at and during 30 days of case hospitalization. Controls were matched to COVID-19 cases based on a combination of age, gender, Charlson Comorbidity Index score (CCI) and index date. Patients aged 18 and older were matched within 10 years and those aged 60 and older within 5 years from individual case. There was a random selection without replacement among eligible controls for each case done. Index date was divided on 3-month periods (until May 2020, June-August 2020, September-November 2020, December-February 2021). The CCI was calculated using medical claims one-year prior to index date containing 16 comorbidities and classified according to Charlson et al [9]. There was a random selection without replacement among eligible controls for each case. We followed all study subjects for up to 12 months starting from 30 days after discharge until the study end (November 15th, 2021), death, diagnosis of the relevant outcome in each analysis or SARS-CoV-2 positive test in control group.

## Outcomes

We aimed to estimate the burden of post-acute COVID-19 sequalae in a cohort of hospitalized SARS-coV-2 positive patients in comparison to infection-free controls from general population. We also explored whether the severity of COVID-19 infection, as proxied by hospitalization and intensive care unit (ICU) admission, affects the risks. The main outcomes were all-cause mortality and all-cause hospital (re)admission. All-cause mortality was defined as death from all causes during study follow-up; all-cause hospital readmission was defined as hospital admission after discharge for COVID-19 for cases, which was compared to hospital admissions in controls. Secondary outcomes were new-onset (incident episodes) of selected clinical conditions identified based on primary ICD-10 codes a (respiratory diseases, heart diseases, stroke, neurological illness, diabetes, chronic kidney, liver disease or mental disorder)

appearing at least once on the medical bill. Table including ICD-10 codes used to define an outcomes are presented in (S1 Table). Incident outcome refers to the occurrence of new case of any of the selected health conditions over a follow-up period for an individual who had not had this condition before (the index date) (recall period of 3 years). Intensive care episode during follow up was captured by admission to intensive care unit on medical bill (health insurance claim).

## Covariates

Covariates for analyses included age, preexisting comorbidities, and indicators of pre-COVID-19 healthcare utilization, such as number of outpatient encounters and number of hospital admissions. Pre-COVID-19 comorbidities were captured using ICD-10 codes for the $3^*365$-day period prior to the index date for case patients and their matched controls from medical bills. Comorbidities were defined as primary or other diagnoses coded at the claim and/or diagnoses of any type on hospital or outpatient health care claims during the three years preceding the index. We applied a restriction to outpatient claims, such that a comorbid condition could be flagged during the preceding period only if it appeared two or more times at least 7 days apart [10]. In the final model we accounted for comorbidities with the prevalence higher than 5% (Table 1).

## Statistical analysis

Distribution of baseline characteristics were compared between cases and controls; and by care type (general ward, ICU) of cases. Descriptive statistics were presented as absolute numbers with proportions or means and standard deviation (SD). The rates of the occurrence of all cause death, readmission, and health outcomes per 10 000 person years in cases and controls were calculated. We estimated 95% confidence intervals (CI) with a Poisson distribution. If the control participant tested positive for SARS-CoV-2 during follow-up they were censored at date of SARS-CoV-2 positive result. The risks of incident outcome were estimated from a Cox survival model. In the survival analysis case-control matching structure was accounted for by including a random effect for each case-controls set in a Cox frailty model. The final Cox survival model was adjusted for comorbidities with prevalence of ≥5%, age, healthcare utilization and previous hospitalization. We produced Kaplan-Meier curves to graphically display the estimates of the significant outcomes. Hazard ratios for each of the outcomes including incident diagnoses were estimated from cause-specific hazard models. All analyses and visualization were done using STATA 17 [11]. To assess heterogeneity related to the matching variables we conducted subgroup analyses stratified by sex and age group (under and over 60 years). This was a retrospective observational cohort study, using claims data. The study was approved by the Research Ethics Committee of the University of Tartu (351/M-8), and no additional requirements were needed regarding gathering informed consent of study participants due to retrospective design of the study.

## Results

The primary cohort comprised 3949 adult patients who were alive 30 days after COVID-19-related hospital admission (24% of them needed ICU during hospitalisation) and 15511 matched individuals from the control cohort. In the final matching process 3760 (95%) cases were matched to 4, 110 to three, 45 to two and 51 to one control with the total of 19460 individuals included in the analysis (3949 cases and 15511 controls). 1183 individuals among the matched controls tested SARS-CoV-2 positive during follow-up period and were censored in survival analyses. The study population demographics and pre-existing health conditions characteristics are presented in Table 1. Both cohorts comprised slightly more women (54.3% cases

**Table 1. Baseline descriptive characteristics of patients who were hospitalized with COVID-19 and matched general population controls during 2020–2021 in Estonia.**

| Characteristics | COVID-19 hospitalized cases n = 3949 | Matched controls n = 15 511 | p value |
|---|---|---|---|
| Age, mean SD;y | 65.4 (16.7) | 64.9 (16.8) | 0.04 |
| Sex; n% female | 2147 (54.3) | 8413 (54.2) | 0.9 |
| Follow up days, mean (SD) | 294.9 (73.6) | 309.6 (46.6) | <0.0001 |
| Charlson comorbidity index 0 1–2 mild 3–4 moderate 5+ severe | 2641(66.8) 1141(28.9) 148 (3.8) 19 (0.5) | 10511 (67.8) 4424 (28.5) 527 (3.4) 49 (0.3) | 0.25 |
| Charlson comorbidity index, mean (SD) | 0.5 (0.9) | 0.5 (0.9) | |
| Hospital admissions | 0.7 (1.6) | 0.3 (0.8) | <0.0001 |
| Outpatient encounters | 10.6 (8.4) | 7.5 (7.5) | <0.0001 |
| **Pre-existing comorbidities**; n (%) | | | |
| Anxiety | 306 (7.8) | 985 (6.4) | 0.002 |
| Chronic kidney disease | 250 (6.3) | 620 (4.0) | <0.0001 |
| Chronic lower respiratory disease | 592 (14.9) | 1613 (10.4) | <0.0001 |
| Chronic liver disease | 116 (2.9) | 306 (1.9) | <0.0001 |
| Dementia | 154 (3.9) | 271 (2.7) | <0.0001 |
| Disorders involving immune mechanism | 18 (0.5) | 27 (0.2) | 0.001 |
| Disorder of lipoprotein | 805 (20.4) | 2727 (17.6) | <0.0001 |
| Diabetes mellitus | | | |
| Type 1 diabetes | 64 (1.6) | 170 (1.0) | 0.007 |
| Type 2 diabetes | 763 (19.3) | 2344 (15.1) | <0.0001 |
| Gastritis and duodenitis | 506 (12.8) | 1456 (9.39) | <0.0001 |
| Guillain-Barré | 2 (0.05) | 6 (0.04) | 0.74 |
| Hypothyroidism | 322 (8.2) | 946 (6.1) | <0.0001 |
| Hypertension | 2539 (64.3) | 8062 (51.9) | <0.0001 |
| Insomnia | 121 (3.1) | 285 (1.8) | <0.0001 |
| Ischemic heart disease | 638 (16.2) | 1991 (12.8) | <0.0001 |
| Mood disorders | 257 (6.5) | 790 (5.1) | <0.0001 |
| Neoplasms | 907 (22.9) | 3091 (19.9) | <0.0001 |
| Obesity | 495 (12.5) | 1001 (6.5) | <0.0001 |
| Other forms of heart diseases | 1100 (27.9) | 3460 (22.3) | <0.0001 |
| Organ transplant | 33 (0.8) | 44 (0.3) | <0.0001 |
| Stroke | 153 (3.9) | 425 (2.7) | <0.0001 |
| Substance abuse | 92 (2.3) | 206 (1.3) | <0.0001 |
| Schizophrenia | 85 (2.2) | 142 (0.9) | <0.0001 |
| Multiple sclerosis | 2 (0.005) | 22 (0.1) | 0.145 |

and 54.2% controls) than men. The mean age of cases was 65.4 (SD 16.7) and in controls 64.9 (SD 16.8) years. The mean follow-up time until death or study end for cases was 294.9 (SD 73.6) and 309.6 (SD 46.6) days for controls. At baseline compared to general population cases were more likely to have a comorbid condition (most notably obesity, hypertension, heart, liver, kidney chronic diseases and psychiatric conditions). Among individuals hospitalized for COVID-19, there were more men among those requiring ICU admission (56.4%). The mean age of ICU patients was 66.7 (SD 14.2) years. In the whole cohort 23.9% of individuals had at least one of outcomes (death, (re)admission, any of the selected health conditions) assessed: 40.3% (95%CI 38.7–41.9) of the cases and 19.4% (95%CI 18.7–19.9) of controls.

## All-cause mortality

During the follow-up period 399 (10%) cases and 561 (3.6%) control group subjects died (Table 2). All-cause mortality was over two times higher among cases with rate per 10 000 person-years of 1332.4 (95% CI 1207.8–1469.7) in comparison to the rate in controls of 467.7 (95%CI 430.6–508.1) (Table 2). Significantly increased risk was observed in cases compared to controls for all-cause mortality aHR 2.57 (95%CI 2.23–2.96) (Fig 1).

Among those required ICU admission 13.1% died during study period (risk of all-cause mortality compared to controls aHR = 3.38 95%CI (2.59–4.40)) (S3 Table).

## All cause hospital (re)admission

Of 3949 cases over the study period 824 (20.9%) were readmitted for any reason. During the follow-up, hospital admission occurred at rate of 3153.9 (95% CI 2945.8–3376.8) for cases and 1482.5 (95% CI 1413.4–1555.0) per 10 000 person-years for controls. We report markedly

**Table 2. Rates of incident outcomes (events per 10 000 person years) among patients who were hospitalized with COVID-19 and a matched general population control in Estonia 2020–2021.**

| Outcome | Group | Person time (x 10 000 person years) | Number of events | Rate (95%CI) per 10 000 person years |
|---|---|---|---|---|
| Death all cause | Cases<br>Controls | 0.29<br>1.19 | 399<br>561 | 1332.4 (1207.8–1469.7)<br>467.7 (430.6–508.1) |
| Readmission all cause | Cases<br>controls | 0.30<br>1.26 | 824<br>1686 | 3153.9 (2945.8–3376.8)<br>1482.5 (1413.4–1555.0) |
| Anxiety | Cases<br>controls | 0.27<br>1.11 | 54<br>118 | 197.7 (151.4–258.1)<br>105.5 (88.1–126.3) |
| Chronic lower respiratory disease | Cases<br>controls | 0.25<br>1.07 | 88<br>96 | 352.1 (285.7–433.9)<br>89.5 (73.3–109.3) |
| Chronic liver disease | Cases<br>controls | 0.29<br>1.18 | 25<br>29 | 86.4 (58.4–127.9)<br>24.7 (17.2–35.5) |
| Chronic kidney disease | Cases<br>controls | 0.28<br>1.15 | 13<br>18 | 46.1 (26.8–79.4)<br>15.6 (9.8–24.8) |
| Disorder of lipoprotein | Cases<br>controls | 0.24<br>0.98 | 40<br>116 | 168.9 (123.9–230.2)<br>117.9 (98.3–141.4) |
| Dementia | Cases<br>controls | 0.29<br>1.18 | 33<br>43 | 114.4 (81.3–160.8)<br>36.5 (27.1–49.2) |
| Gastritis and duodenitis | Cases<br>controls | 0.26<br>1.08 | 85<br>165 | 329.7 (266.6–407.8)<br>152.5 (130.9–177.6) |
| Hypertension | Case<br>controls | 0.10<br>0.57 | 111<br>251 | 1058.8 (879.1–1275.3)<br>438.7 (387.7–496.5) |
| Hypothyroidism | Cases<br>Controls | 0.27<br>1.13 | 15<br>32 | 54.7 (32.9–90.8)<br>28.4 (20.1–40.2) |
| Insomnia | Cases<br>controls | 0.29<br>1.18 | 25<br>42 | 86.3 (58.3–127.7)<br>35.7 (26.4–48.3) |
| Ischemic heart disease | Cases<br>controls | 0.25<br>1.04 | 107<br>146 | 431.6 (357.1–521.6)<br>139.9 (118.9–164.6) |
| Mood disorders | Cases<br>controls | 0.28<br>1.13 | 46<br>78 | 165.3 (123.8–220.7)<br>68.8 (55.1–85.8) |
| Other forms of heart disease | Cases<br>controls | 0.21<br>0.93 | 158<br>191 | 749.1 (640.9–875.5)<br>205.1 (177.9–236.3) |
| Stroke | Cases<br>controls | 0.29<br>1.16 | 50<br>108 | 174.9 (132.6–230.9)<br>92.8 (76.8–112.0) |
| Substance abuse | Cases<br>controls | 0.29<br>1.18 | 11<br>28 | 37.58 (20.81–67.86)<br>23.68 (16.35–34.29) |
| Type 2 diabetes | Cases<br>controls | 0.24<br>1.02 | 61<br>75 | 254.9 (198.4–327.7)<br>73.6 (58.7–92.3) |

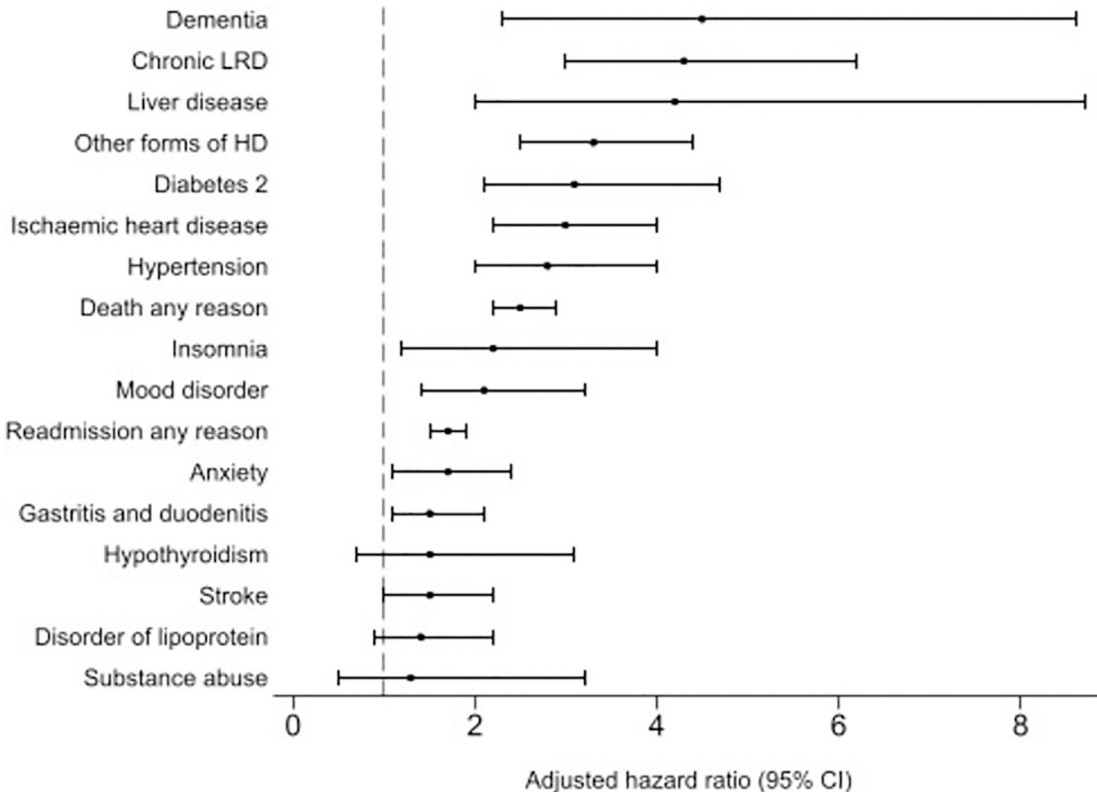

**Fig 1. Adjusted HRs comparing COVID-19 hospitalised patients and controls for risk of post-acute COVID-19 sequalae in Estonia 2020–2021.** Circles represent the hazard ratio, and the horizontal bars extend from the lower limit to the upper limit of the 95% confidence interval. Abbreviations LRD lower respiratory diseases; HD heart diseases.

increased risk of all cause readmission aHR 1.73 (95%CI 1.58–1.90) in cases compared to age, sex and CCI matched controls.

## Cases of new selected clinical sequelae

The data on incident cases of clinical sequelae are provided in Table 2. The occurrence of newly diagnosed hypothyroidism (p = 0.13), disorder of lipoprotein (p = 0.06), and substance abuse (p = 0.42) did not differ between cases and controls. Higher risk for all other clinical sequelae was observed in COVID-19 hospitalized patients compared to the matched general population group.

After adjustment for age, healthcare utilization, previous hospitalization and 11 clinical sequelae cases had significantly increased risk for all outcomes except hypothyroidism, substance abuse and disorder of lipoprotein metabolism compared to matched general population group (Figs 1 and 2 and S2 Table).

The largest increases in risk were observed for dementia aHR = 4.50 (95% CI 2.35–8.64), chronic lower respiratory disease aHR = 4.39 (95% CI 3.09–6.22), liver disease aHR 4.20 (95% CI 2.01–8.77) and other (than ischemic) forms of heart diseases (ICD-10 I30-52) aHR = 3.39 (95%CI 2.58–4.44). Chronic kidney disease was not included in the final model due to a small number of cases.

## Risks of outcomes across subgroups

To assess heterogeneity related to a matching variable we conducted subgroup analysis. Significantly increased risk was universal and was observed for the majority of adverse events

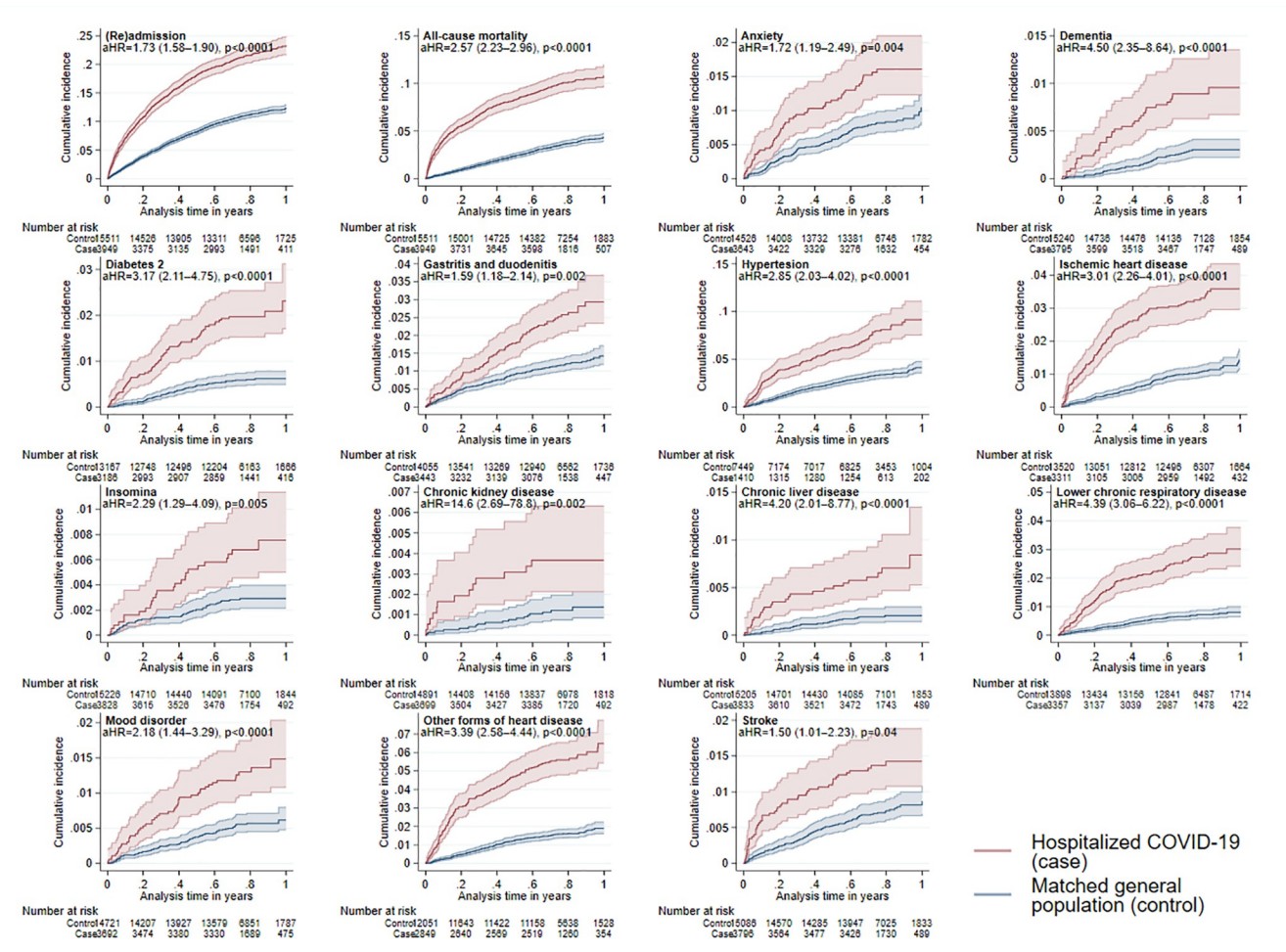

**Fig 2. Kaplan-Meier curves for the comparison of the significant outcomes during 12 months comparing hospitalized COVID-19 patients (red curves) and matched general population (blue curves).** Shaded zones indicate 95% CI.

across both age and gender groups. Biggest differences in hazard ratios were observed for all-cause mortality in individuals aged <60 years aHR 5.99 (95%CI 3.45–10.38) versus aged ≥60 years aHR 2.76 (95%CI 2.42–3.15) and readmission <60 years with aHR 2.72 (95% CI 2.27–3.27) versus aged ≥60 years aHR 2.0 (1.82–2.19). No substantial differences in aHRs for cases vs controls were observed between males and females across the outcomes analysed.

## Discussion

We conducted a comprehensive study investigating long term health complications after an acute COVID-19 requiring hospitalization. We saw excess mortality, hospital readmission and substantial risk for the emergence of several chronic diseases among a whole country's population of adults hospitalized for COVID-19 compared with age, gender and CCImatched general population. The impact of severe COVID-19 was further accentuated among those requiring ICU. One of the main strengths of our study stems from long follow-up after the initial COVID-19 episode [12].

## All-cause mortality and re-admission

Rates for all cause readmission (21%) and death (10%) reported in our study are consistent with the Donelly et al. study from USA, which reported rates within 60 days of discharge of these outcomes prior to mass vaccination of 20% and 9% respectively among COVID-19 hospitalization survivors [13]. Patients hospitalised due to COVID-19 and surviving at least 30 days had more than 2.5-fold higher risk of all-cause mortality and almost double risk for subsequent hospitalization than controls from general population. Our findings of relative risks are consistent with study results from England showing that the overall risk of hospitalisation or death was higher in the COVID-19 group than general population controls [14].

The severity of COVID-19 had a clear effect on all-cause mortality, readmission and clinical conditions, risks were higher in those COVID-19 patients requiring ICU admission. The inclusion of COVID-19 patients requiring ICU in our analysis is important because the current literature is limited regarding long term complications in those patients.

We found that 40.3% of individuals hospitalized due to COVID-19 developed at least one new clinical condition with in one year after the acute phase of SARS-CoV-2 infection, and in comparison, to age/gender and CCI matched peers, this is at least doubling of the risk.

## New clinical conditions after acute COVID-19

Individuals surviving after COVID-19 hospitalization are at increased risk for several new clinical conditions; especially for chronic lower respiratory disease, other forms of heart diseases (ICD-10 I30-52), dementia and chronic liver disease.

In our study the largest increase in risk of 4.5 times was observed for dementia. Adequate post-discharge care is vital for those vulnerable group. Two-fold higher risk for neuropsychiatric sequelae as mood disorder and anxiety in our study is in line with the results reported in previously mentioned studies investigating neurological and psychiatric outcomes in COVID-19 hospitalized individuals [15].

Increased risk for chronic lower respiratory diseases can be explained by the finding from China reporting restrictive pulmonary physiology in COVID-19 hospitalized patients [16]. A study from USA with shorter follow-up reported that 65% of severe cases survivors experience fibrotic changes after COVID-19 [17]. Incidence of pulmonary sequelae is the aftermath of both virus-dependent (invasion and alveolar damage by the virus) and virus-independent mechanisms that cause persistent lung damage [18].

Furthermore, we report risk for chronic liver disease was four times higher in hospitalized COVID-19 patients, which is supported by the results reported in the study from UK [19]. In our study the risk of incident type 2 diabetes among COVID-19 patients of being 3 times higher than in general population cohort which is similar to that reported previously [20]. Al-Aly et al reported elevated risk of other cardiovascular disorders compared to general population without evidence of COVID-19 [21].

Our results are in line with reported evidence of biological mechanisms associated with respiratory [22], cardiovascular [23], metabolic [24] and neuropsychiatric [15] post-acute COVID-19 sequelae. Mechanisms surrounding post-acute COVID-19 complications are not entirely clear, but direct effects including with long-term tissue damage or strong immune system reaction leading to autoimmunity or immune dysregulation [25]; direct viral effect (viral invasion affecting heart function [26]; along with indirect effects as loneliness, isolation, economic situation has been described [27,28].

It is important to put these findings into the context of long-term effects of other infectious (viral) diseases as COVID-19 has brought to attention previously understudied phenomenon of post-acute infection syndrome. Post-infectious sequelae have been documented following

infection by other viral infections (Ebola, influenza, dengue) including coronaviruses. Studies conducted up to 15 years reported effects on respiratory system [29], mental health [30], lipid metabolism [31] after acute SARS. In the population-based study from England most risks were similar to those observed after influenza hospitalizations, but COVID-19 patients had higher risks of all-cause mortality, readmission due to the initial infection and dementia, highlighting the importance of post-discharge monitoring [14]. A study from China has shown that patients who survive influenza A (H7N9) virus infection are at risk of physical and psychological complications of lung injury and multi-organ dysfunction [32].

To inform prevention and treatment strategy persistent follow-up of survivors is necessary. More emphasis has to be put on those experiencing severe initial COVID-19, as study from Denmark found, that the risk of severe post-acute sequelae after SARS-CoV-2 infection without hospital admission is low [33].

## Strengths and limitations

To our knowledge this is one of the largest cohort studies with the longest follow-up time assessing the post-acute COVID-19 consequences in adult COVID-19 hospitalised patients in Europe. We believe our findings are robust given the sample size, the age, sex and CCI matching and the completeness of the data. Using nationwide data and long term follow-up allowed us to obtain precise estimates of assessed outcomes.

However, the study does have some limitations. Due to the nature of study design, we lack important information on subjective symptoms and other self-reported health characteristics that can be obtained from COVID-19 survivors (fatigue, poor attention, altered sense of taste). Accordingly, we are potentially missing a subset of post-COVID-19 health problems. Outcomes were presented in broad categories and more specific and granular presentation would be of a future interest.

Furthermore, there could be a number of patients with unrecorded post-acute COVID-19 features due to health care service provision limitation making our incidence estimates lower. It is difficult to gauge the completeness of electronic health records, or to validate diagnoses, and these factors could influence incidence estimates with the risk of measurement bias. We hypothesise that these deficiencies would affect cases and controls similarly leading to non-differential misclassification. The hospital admission thresholds could be lower in post-COVID-19 individuals than in the general population, leading to an overestimation of the impact on health. Due to lockdowns and inadequate access to medical services, rates of diagnoses might have decreased in individuals without hospital admission, which could lead to an underestimation of associations. Also, there could be individuals among the matched general population cohort that were untested for COVID-19, leading to an underestimation of effects. Based on the nature of the study design we were unable to match for all potentially relevant factors [34], for example, we did not have data on sociodemographic factors. Moreover, we showed that COVID-19 hospitalized patients were more likely to have baseline comorbidities than the general population controls, reflecting known associations between comorbidities and risks of severe COVID-19 outcomes. We have accounted for this by matching our cohorts for age, sex, Charlson Comorbidity Index and adjusting the final Cox survival model for a rich set of covariates to remove confounding as imbalance existed after matching. The adjustment was made for preexisting comorbidities with a prevalence of ≥5%, health care utilization, and previous hospitalization. We do agree that the propensity score matching would be also another appropriate way to balance exposed groups on the confounding factors. In addition, we do not know which SARS-CoV-2 virus-type individuals were infected. Wild type was overwhelmingly prevalent in Estonia during the current study period [35] It has been reported that some

SARS-CoV-2 variant types are overrepresented [36] in severe cases and could possibly modify the incidence of post-acute COVID-19 manifestations [37].

Regarding the true impact of SARS-CoV-2, in medical literature the term of post-intensive care syndrome (PICS) has been described [38]. PICS manifestation involves physical, mental and cognitive health impairments after ICU discharge, which are partially similar to post-acute COVID-19 sequelae. In this regard, more studies are required in order to separate the impact of PICS and a true effect of SARS-CoV-2 leading to the consequences after acute phase of COVID-19. Finally, observational epidemiological studies cannot identify mechanisms of effect and causality; complementary studies are needed.

It is important to note that the current study was conducted before mass vaccination was introduced. There are a limited number of studies looking at whether vaccination reduces the risk of post-acute COVID-19 clinical complications. In the large cohort study by Taquet et al. analysis showed that the risk of several important outcomes including diabetes, mood disorder, anxiety appear to be unaffected by vaccination suggesting that other biological mechanisms support those outcomes [39]. More nationwide studies with standardized design, longer follow-up time and involving vaccinated individuals are therefore needed.

## Conclusion

In summary the present study shows that after the acute-phase of COVID-19 hospitalized patients are at significantly higher risk of death and readmission, as well as cardiovascular, respiratory and neuropsychiatric outcomes over the next 12 months. This information could help with service planning for those with post-acute COVID-19 sequelae. With more than 17 000 cases hospitalised due to COVID-19 in Estonia by May 2022, our results suggests that the burden of COVID-19 related comorbidity on healthcare system will be substantial. Our results highlight those needed hospitalisation during acute phase of COVID-19 are in need of post-discharge follow up and multi-disciplinary care. Further efforts are needed to describe full spectrum of post-acute COVID-19 sequelae and to investigate the impact of PICS and vaccination on long-term sequelae of SARS-CoV-2 infection.

## Supporting information

**S1 Table. Outcomes measured and definitions.**
(RTF)

**S2 Table. Hazard ratios (HR) for major incident post-acute COVID-19 sequelae in hospitalised patients and matched general population controls in Estonia 2020–2021.**
(RTF)

**S3 Table. Hazard ratios (HR) for major incident post-acute COVID-19 sequelae in patients admitted to intensive care unit and matched general population controls in Estonia 2020–2021.**
(RTF)

## Author Contributions

**Conceptualization:** Anna Tisler, Oliver Stirrup, Heti Pisarev, Anneli Uusküla.

**Data curation:** Anna Tisler, Heti Pisarev, Ruth Kalda, Marko Piirsoo, Anneli Uusküla.

**Formal analysis:** Anna Tisler.

**Funding acquisition:** Ruth Kalda, Tatjana Meister, Kadri Suija, Raivo Kolde, Marko Piirsoo, Anneli Uusküla.

**Investigation:** Anneli Uusküla.

**Methodology:** Anna Tisler, Oliver Stirrup, Heti Pisarev, Anneli Uusküla.

**Project administration:** Anna Tisler, Oliver Stirrup, Ruth Kalda, Kadri Suija, Raivo Kolde.

**Resources:** Anna Tisler, Tatjana Meister, Kadri Suija, Raivo Kolde, Marko Piirsoo, Anneli Uusküla.

**Software:** Anna Tisler.

**Supervision:** Anna Tisler, Oliver Stirrup, Anneli Uusküla.

**Validation:** Anna Tisler, Oliver Stirrup, Heti Pisarev, Tatjana Meister, Kadri Suija, Raivo Kolde, Marko Piirsoo, Anneli Uusküla.

**Visualization:** Anna Tisler, Oliver Stirrup, Heti Pisarev.

**Writing – original draft:** Anna Tisler, Oliver Stirrup, Anneli Uusküla.

**Writing – review & editing:** Anna Tisler, Oliver Stirrup, Heti Pisarev, Ruth Kalda, Tatjana Meister, Kadri Suija, Raivo Kolde, Marko Piirsoo, Anneli Uusküla.

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
