## [Decision Letter · Decision Letter 0]

1 Aug 2022

PONE-D-22-19899Post-acute sequelae of COVID-19 among hospitalized patients in Estonia: nationwide matched cohort studyPLOS ONE

Dear Dr. tisler,

Thank you for submitting your manuscript to PLOS ONE. After careful consideration, we feel that it has merit but does not fully meet PLOS ONE’s publication criteria as it currently stands. Therefore, we invite you to submit a revised version of the manuscript that addresses the points raised during the review process.

In addition to the reviewers' comments and suggestions, more clarifications should be written regarding the methods and diagnostic codes (ICD-10) using 1 or 2 codes to confirm the diagnosis of diseases.

We look forward to receiving your revised manuscript.

Kind regards,

Aqeel M Alenazi

Academic Editor

PLOS ONE

Journal Requirements:

Additional Editor Comments:

Reviewers indicated limitations in the selection of the cohorts and unbalanced diseases with some suggestions to improve such as propensity score matching. In addition, more clarifications should be written regarding the methods and diagnostic codes (ICD-10). For example, have you used one or 2 codes to confirm the diagnosis of diseases? This will improve the validity and accuracy of the results.

Reviewers' comments:

Reviewer's Responses to Questions

**Comments to the Author**

1. Is the manuscript technically sound, and do the data support the conclusions?

Reviewer #1: Yes

Reviewer #2: Partly

2. Has the statistical analysis been performed appropriately and rigorously? 

Reviewer #1: I Don't Know

Reviewer #2: Yes

3. Have the authors made all data underlying the findings in their manuscript fully available?

Reviewer #1: Yes

Reviewer #2: Yes

4. Is the manuscript presented in an intelligible fashion and written in standard English?

Reviewer #1: Yes

Reviewer #2: Yes

5. Review Comments to the Author

Reviewer #1: The paper submitted by Anna Tisler et al analyzes clinical outcomes of patients hospitalized and surviving after COVID-19 disease in a nationwide study (Estonia) during the pre-vaccine era. They used a matched control population by age, sex, and Charlson’s comorbidity index (CCI). This kind of studies are relevant for health systems to design policies of surveillance after patients are discharged. The main objectives and the methods employed are clearly described in the manuscript. However, I have some concerns to address to the authors:

1. Despite cases and controls were matched by age, sex and CCI, the prevalence of relevant comorbidities was higher in the cohort of patients with COVID-19 (e.g., arterial hypertension, neoplasms, type 2 diabetes, ischemic and non-ischemic heart disease, etc). I wonder whether it is possible to do another approach taking into consideration for matching conditions with a high prevalence (for example, > 5%) in both cohorts.

2. In table 2 it is presented the incidence of outcomes among COVID-19 and control patients. Two different measures are presented: person-time and rate (both by 10.000 person-years). Despite I cannot repeat your calculations, rate was higher in COVID-19 patients than in controls (for example, in the case of death from all cause the figures were 1332 and 467, respectively). Conversely, when the same results are presented as person time the figure is higher in controls (1.19 vs. 0.29 for all cause mortality). Can you explain me this discrepancy?

Reviewer #2: Authors wanted to examine post-acute sequelae of Covid-19 disease with a matched cohort study in Estonia. Data came from nationwide healthcare registry. Two patient groups were matched according to age, gender, and CCI score; Covid-19 and others. Although, these three features (age, gender and CCI score) were used for matching, it is obvious that Covid-19 patients had higher rate of pre-existing comorbidities than those of counter parts. So, it is expected that Covid-19 patients could have higher rates of hazard ratios or mortality, and results could be overestimated. Propensity score matching could be more appropriate to reduce these differences in the analysis. If it is not possible to do this test, this limitation should be added to the paper.

Other concern is that how does this paper give contribution to the literature, as nearly all the population was vaccinated in Europe. Also, we still don’t know whether vaccination changed long term disease effects. Or, how long time should these patients will be watched for sequelae? These are topics that should be discussed in the paper.

Other recommendations;

Line 135: “all-cause” rather than “all cause”.

Line 189: It is stated that “At baseline compared to general population cases were more likely to have a comorbid condition (most notably obesity …” Is there any information regarding obesity in the paper?

Discussion;

It was used that “age, gender and co-morbidities matched general population.” in a few sentences. Actually, it wasn’t a fully matched cohort. It can be used “CCI score” rather than “co-morbidities”.

Abstract;

The results may not provide “robust” estimates for post disease morbidity because the groups were not fully matched. This statement needs revision.

6. PLOS authors have the option to publish the peer review history of their article (what does this mean?). If published, this will include your full peer review and any attached files.

Reviewer #1: **Yes: **Francesc Moreso

Reviewer #2: **Yes: **Mehmet Ali Aslaner

---

## [Author Response · Author response to Decision Letter 0]

15 Sep 2022

RESPONSE TO REVIEWERS 

General comment:

In addition to the reviewers' comments and suggestions, more clarifications should be written regarding the methods and diagnostic codes (ICD-10) using 1 or 2 codes to confirm the diagnosis of diseases.

Response: Thank you for the comment. We have edited Methods section to clarify the use of ICD-10 codes.

Reviewer #1:

1. Despite cases and controls were matched by age, sex and CCI, the prevalence of relevant comorbidities was higher in the cohort of patients with COVID-19 (e.g., arterial hypertension, neoplasms, type 2 diabetes, ischemic and non-ischemic heart disease, etc). I wonder whether it is possible to do another approach taking into consideration for matching conditions with a high prevalence (for example, > 5%) in both cohorts.

Response: Thank you for pointing that out.

In the addition to the matching for age, sex, Charlson Comorbidity Index the final Cox survival model was adjusted for comorbidities with prevalence of ≥5%, health care utilization and previous hospitalization that we believe removed confounding if imbalance existed after matching.

2. In table 2 it is presented the incidence of outcomes among COVID-19 and control patients. Two different measures are presented: person-time and rate (both by 10.000 person-years). Despite I cannot repeat your calculations, rate was higher in COVID-19 patients than in controls (for example, in the case of death from all cause the figures were 1332 and 467, respectively). Conversely, when the same results are presented as person time the figure is higher in controls (1.19 vs. 0.29 for all cause mortality). Can you explain me this discrepancy?

Response: Cases were matched to a random subset of individuals insured by the EHIF who at the date of the first positive SARS-CoV-2 test of the COVID-19 case were alive and had no evidence of COVID-19 using a case to control ratio of 1:4 and that is the reason for person time being 4 times higher for controls (controls 1.19 versus cases 0.29 x 10 000 person years).

Reviewer #2:

1. Authors wanted to examine post-acute sequelae of Covid-19 disease with a matched cohort study in Estonia. Data came from nationwide healthcare registry. Two patient groups were matched according to age, gender, and CCI score; Covid-19 and others. Although, these three features (age, gender and CCI score) were used for matching, it is obvious that Covid-19 patients had higher rate of pre-existing comorbidities than those of counter parts. So, it is expected that Covid-19 patients could have higher rates of hazard ratios or mortality, and results could be overestimated. Propensity score matching could be more appropriate to reduce these differences in the analysis. If it is not possible to do this test, this limitation should be added to the paper.

Thank you for your comment, we do agree that the propensity score matching would be also another appropriate way to balance exposed groups on the confounding factors, but we believe that matching for for age, sex, Charlson Comorbidity Index and adjusting final Cox survival model for comorbidities with prevalence of ≥5%, health care utilization and previous hospitalization has removed confounding as imbalance existed after matching.

2. Other concern is that how does this paper give contribution to the literature, as nearly all the population was vaccinated in Europe. Also, we still don’t know whether vaccination changed long term disease effects. Or, how long time should these patients will be watched for sequelae? These are topics that should be discussed in the paper.

Our findings are consistent with emerging evidence from early studies suggesting that a subset of people infected with SARS-CoV-2 can experience health problems several months after the acute phase of their infection; however, epidemiological characterisation of such sequelae has been limited. Only a few other studies to date have compared post-COVID risks with a control group in using nationwide data.

The present study helps to contextualise these observations by adding explicit comparison with risks experienced by the general population. In addition we believe that our results can be generalised to those countries with similar population structure with lower vaccination coverage.

We also believe that our results are relevant in terms of future and present pandemics as we note in the manuscript that the post-infectious sequelae have been documented following infection by other viral infections (Ebola, influenza, dengue) and only 63.2% of European population is fully vaccinated 1.

1 https://ourworldindata.org/covid-vaccinations?country=OWID_WRL

Ongoing monitoring will be important to investigate whether these patterns persist in the light of new variants and increasing levels of vaccination.

3. Line 135: “all-cause” rather than “all cause”

Response: corrected

4. Line 189: It is stated that “At baseline compared to general population cases were more likely to have a comorbid condition (most notably obesity …” Is there any information regarding obesity in the paper?

Response: Thank you for pointing that out. We have now included it in the Tabel 1, we double- checked and it was already included in the final model.

5. It was used that “age, gender and co-morbidities matched general population.” in a few sentences. Actually, it wasn’t a fully matched cohort. It can be used “CCI score” rather than “co-morbidities”

Response: corrected

6. The results may not provide “robust” estimates for post disease morbidity because the groups were not fully matched. This statement needs revision.

Response: The reviewed statement is now included.

---

## [Decision Letter · Decision Letter 1]

18 Oct 2022

PONE-D-22-19899R1Post-acute sequelae of COVID-19 among hospitalized patients in Estonia: nationwide matched cohort studyPLOS ONE

Dear Dr. tisler,

Thank you for submitting your manuscript to PLOS ONE. After careful consideration, we feel that it has merit but does not fully meet PLOS ONE’s publication criteria as it currently stands. Therefore, we invite you to submit a revised version of the manuscript that addresses the points raised during the review process.

Please, revise your manuscript or provide explanation to the reviewer's comments

We look forward to receiving your revised manuscript.

Kind regards,

Aqeel M Alenazi

Academic Editor

PLOS ONE

Journal Requirements:

Additional Editor Comments:

Please, revise your manuscript or provide explanation to the reviewer's comments

Reviewers' comments:

Reviewer's Responses to Questions

**Comments to the Author**

1. If the authors have adequately addressed your comments raised in a previous round of review and you feel that this manuscript is now acceptable for publication, you may indicate that here to bypass the “Comments to the Author” section, enter your conflict of interest statement in the “Confidential to Editor” section, and submit your "Accept" recommendation.

Reviewer #1: All comments have been addressed

Reviewer #2: (No Response)

2. Is the manuscript technically sound, and do the data support the conclusions?

Reviewer #1: Yes

Reviewer #2: (No Response)

3. Has the statistical analysis been performed appropriately and rigorously? 

Reviewer #1: I Don't Know

Reviewer #2: (No Response)

4. Have the authors made all data underlying the findings in their manuscript fully available?

Reviewer #1: Yes

Reviewer #2: (No Response)

5. Is the manuscript presented in an intelligible fashion and written in standard English?

Reviewer #1: Yes

Reviewer #2: (No Response)

6. Review Comments to the Author

Reviewer #1: The limitation that both reviewers have raised about the matching procedure of both cohorts is not fully addressed. Is it possible to match both populations by the baseline comorbidities (for example by a propensity socre)?

Reviewer #2: (No Response)

7. PLOS authors have the option to publish the peer review history of their article (what does this mean?). If published, this will include your full peer review and any attached files.

Reviewer #1: **Yes: **Francesc Moreso

Reviewer #2: **Yes: **Mehmet Ali Aslaner

---

## [Author Response · Author response to Decision Letter 1]

25 Oct 2022

RESPONSE TO REVIEWERS 

Reviewer #1: 

The limitation that both reviewers have raised about the matching procedure of both cohorts is not fully addressed. Is it possible to match both populations by the baseline comorbidities (for example by a propensity score)?

Response: We are sorry that the matching procedure was not presented clearly. In the analysis presented both populations are matched by the baseline comorbidities burden (Charlson Comorbidity Index). 

In our analysis, we matched our cohorts for age, sex, and Charlson Comorbidity Index and adjusted the final Cox survival model for a rich set of covariates to remove confounding as imbalance existed after matching. Adjustment for preexisting comorbidities with a prevalence of ≥5%, health care utilization, and previous hospitalization was made. We believe that this was sufficient to balance the cohorts out. We do agree with the reviewer, that the propensity score matching would be also another appropriate way to balance exposed groups on the confounding factors.

We have addressed this comment by adding an explanation in the Discussion section on page 20 line 359.

---

## [Decision Letter · Decision Letter 2]

9 Nov 2022

Post-acute sequelae of COVID-19 among hospitalized patients in Estonia: nationwide matched cohort study

PONE-D-22-19899R2

Dear Dr. tisler,

We’re pleased to inform you that your manuscript has been judged scientifically suitable for publication and will be formally accepted for publication once it meets all outstanding technical requirements.

Kind regards,

Aqeel M Alenazi

Academic Editor

PLOS ONE

Additional Editor Comments (optional):

The authors addressed all comments from the reviewers.

Reviewers' comments:

Reviewer's Responses to Questions

**Comments to the Author**

1. If the authors have adequately addressed your comments raised in a previous round of review and you feel that this manuscript is now acceptable for publication, you may indicate that here to bypass the “Comments to the Author” section, enter your conflict of interest statement in the “Confidential to Editor” section, and submit your "Accept" recommendation.

Reviewer #1: All comments have been addressed

Reviewer #2: All comments have been addressed

2. Is the manuscript technically sound, and do the data support the conclusions?

Reviewer #1: Yes

Reviewer #2: (No Response)

3. Has the statistical analysis been performed appropriately and rigorously? 

Reviewer #1: I Don't Know

Reviewer #2: (No Response)

4. Have the authors made all data underlying the findings in their manuscript fully available?

Reviewer #1: No

Reviewer #2: (No Response)

5. Is the manuscript presented in an intelligible fashion and written in standard English?

Reviewer #1: Yes

Reviewer #2: (No Response)

6. Review Comments to the Author

Reviewer #1: The authors have answer my concerns appropriately. I suggest to the Editor to ask for an statistician to evaluate whether the analysis done in the present paper are correct.

Reviewer #2: (No Response)

7. PLOS authors have the option to publish the peer review history of their article (what does this mean?). If published, this will include your full peer review and any attached files.

Reviewer #1: **Yes: **Francesc Moreso

Reviewer #2: **Yes: **Mehmet Ali Aslaner

---

## [Editor Report · Acceptance letter]

14 Nov 2022

PONE-D-22-19899R2 

Post-acute sequelae of COVID-19 among hospitalized patients in Estonia: nationwide matched cohort study 

Dear Dr. Tisler:

I'm pleased to inform you that your manuscript has been deemed suitable for publication in PLOS ONE. Congratulations! Your manuscript is now with our production department. 

Kind regards, 

on behalf of

Dr. Aqeel M Alenazi 

Academic Editor

PLOS ONE